# Agroforestry: Challenges and Opportunities in Rhino Camp and Imvepi Refugee Settlements of Arua District, Northern Uganda

Evangeline Grosrenaud [1,*], Clement Akais Okia [2], Andrew Adam-Bradford [1] and Liz Trenchard [1]

[1] Centre for Agroecology, Water and Resilience, Coventry University, Coventry CV8 3LG, UK; andrewadambradford@gmail.com (A.A.-B.); apy157@coventry.ac.uk (L.T.)
[2] Uganda Country Representative—World Agroforestry Centre, World Agroforestry Centre (Uganda Country Office), Kampala 26416, Uganda; c.okia@cgiar.org
[*] Correspondence: grosrene@uni.coventry.ac.uk

**Abstract:** In the past, the environment has been a low priority in humanitarian operations for refugee agencies and implementing partners because of the emergency context. However, actions to safeguard the environment can be undertaken concurrently with emergency interventions and organisations should take responsibility for conserving the environment in refugee settlements in the same way that they are responsible for the welfare of refugees. Tree-based interventions, such as agroforestry, have been demonstrated as a viable option for resilience and sustainability in landscapes with increasing human pressure. Refugee settlements are subject to intense human pressure and suffer environmental degradation as a consequence. The potential benefits of agroforestry in refugee settlements though are not well researched. This study explores the implementation of agroforestry schemes in refugee settlements in the Arua district of Uganda. Using semi-structured interviews with the beneficiaries of the International Centre for Research on Agroforestry (ICRAF) agroforestry projects in Imvepi and Rhino camps, the study identifies key benefits for participants and the environment. These include improved livelihoods and nutrition. However, there are challenges to overcome before agroforestry can be made more widely available in refugee camps. Key barriers include insufficient land, limited water availability and lack of local knowledge, which limits productivity. This research shows how relief, rehabilitation and development can work hand in hand to reduce social and environmental pressure in the targeted refugee settlements and host communities and improve the well-being of beneficiary households by creating opportunities for income generation, improving nutrition and contributing to social cohesion.

**Keywords:** agroforestry; multifunctional landscapes; resilience; relief; restoration; environment

## 1. Introduction

> *"Agroforestry is the deliberate integration and management of trees on farms and in landscapes"* [1].

Agroforestry is a sustainable resource management system that deliberately associates woody perennials with annual crops and/or livestock in systems that are ecologically adapted to the local soil, water and climatic conditions and which support local biota [2,3]. In other words, it is a system where trees interact with agriculture and/or livestock. Interaction can occur at the landscape or farm level [4]. Agroforestry systems are complex in comparison to monocultures typically found in intensive systems because they usually comprise of different species of annual and perennial plants integrated with livestock. This complexity can help buffer a variety of shocks and stresses at all levels, supporting ecological biodiversity and the economic welfare of the farmer [5,6].

Trees can be incorporated into agricultural systems in several ways. Agroforestry in linear arrangements means that trees are planted on boundaries as live fences or in timber

belts to create windbreaks. They can be also be planted on contours in terraces or bunds as well as used to form barrier hedges. Planting on contours also reduces soil erosion and increases surface water infiltration. Other linear systems include alley cropping in which arable or horticultural crops are grown between the rows of trees. Non-linear agroforestry systems include silvopastoral systems where trees can be clumped or in blocks; these are often used as fodder banks or on farm woodlands [7]. Animals can also be allowed to graze or browse in managed wooded or forested land. Another type of system consists of crops planted under existing tree cover. This can be done underneath scattered or shade trees in cropland and parklands, in orchards, and in plantation crop combinations. Furthermore, agroforestry is a system that is often used in home gardens, village forest gardens and mixed woodlots. Finally, there are sequential agroforestry systems, and these include shifting cultivation, tree improved fallows and taungya [8].

In each of these different types of agroforestry systems, the trees form part of a multifunctional landscape, providing a wide range of ecosystem services. These include provisioning services such as food for humans and animals, wood for construction and fuel, and sources of medicines. In many instances, trees that are multipurpose, those which can provide a range of provisioning services are preferred [9]. Trees also provide many regulating services, improving local and global air quality, enhancing soil fertility and water availability. Incorporating trees and shrubs into an agroforestry system can also improve habitat providing a greater mosaic of vegetation increasing biodiversity and other supporting services, for example, natural pest control. Traditional agroforestry systems may also provide cultural ecosystem services, such as recreation, support well-being and provide a sense of place [10,11].

Agroforestry systems are most effectively implemented when they are adapted to local settings and consider social, economic and cultural aspects. For the poorest not to be left behind when implementing agroforestry projects, it is important to take into consideration the local milieu and to understand how the farmers operate [12]. If implemented appropriately, agroforestry systems have the potential to improve the resilience and adaptive capacity of local farming systems and provide livelihood opportunities for the farmers [11,13,14]. Tumwebaze and Byakagaba also argue that agroforestry is able to enhance food security and augment household income [15]. This is mostly because agroforestry provides a diversification of income. Trees are an important source of food and can play a critical role in communities who suffer from food insecurity and malnutrition [16]. They provide nutrition directly through the supply of nuts and fruits, but they can also assist in putting food on the table in several different ways. Indirect support comes in various forms including; fuel wood, timber, pesticides, and fodder [11]. The bark and leaves of some trees can be used for medicinal purposes, e.g., in laboratory tests neem extracts have shown potential both as a treatment for malaria and can also be used to kill mosquito larvae [17,18]. Trees also provide timber that can be used as building material or for crafts. Indeed, agroforestry trees can produce a wide range of other products that include oils, resins, tannins, pigments, latex, mushrooms, fibres, wax, and honey, and for this reason they have the capacity to diversify income at different times of the year and in the long term [11,19]. Income generated from these activities can make a significant contribution for households that are food insecure because of low employment opportunities [20]. In most agroforestry systems, much of the land surrounding the trees planted remains available for crop production [21]. In fact, trees in agroforestry systems are often multifunctional, for example, nitrogen-fixing trees can improve soil fertility and this in turn can increase food security in the local region by increasing food production. Thus, it is possible to enhance resilience and sustainability of smallholder agriculture through adapted agroforestry systems that can reinforce a diversification of income at different times of the year.

### 1.1. The Potential for Agroforestry in a Refugee Setting

People depend on trees for various reasons; in African regions some of the most important services provided by trees are wood, shade and soil fertility [19,22]. Despite the opportunities that agroforestry offers, there are numerous sets of challenges involved with implementing agroforestry systems in refugee settings. This article assesses those opportunities and challenges and provides recommendations on how to overcome issues and to which extent opportunities can benefit refugees and host communities.

In Uganda, refugee settlements are developed on land provided by the Ugandan government. Normally, refugee households are offered a parcel of land to enable them to build a home, cultivate some food and be more independent—usually parcels of land are around 30 × 30 m. When refugees have limited access to land or attempt to expand their production, they sometimes manage to make arrangements with local communities or individuals to access additional land for the purpose of securing extra food or to generate income. In most situations, refugee agriculture is small-scale, spontaneous, low input, and based on traditional experience and consequently is often not adapted to local settings. Moreover, it is frequently unsustainable, and refugees find themselves with land that does not produce sufficient yields over successive years. This can be the case even in situations where sufficient land is allocated to refugees because it is not managed in a sustainable way and with a long-term vision [23].

In an emergency context or in a protracted emergency, it is difficult to maintain a sustainable lifestyle. Refugees need to eat three times a day, which means that each household will be using fuelwood about 2–3 times a day. When they first arrive, refugees tend to cut down branches to cook and fell trees to build semi-permanent houses. This impacts the environment in that the tree cover in the settlement reduces at a rapid pace and where programmes have been implemented to restore the environment these have been ineffective.

Therefore, the International Centre for Research on Agroforestry (ICRAF) started a tree nursery and tree planting programme in Imvepi and Rhino Camp settlements in the Arua district of Uganda. The ICRAF project aims to raise tree seedlings that are adapted to the local environment, favouring local species and distributing the seedlings for free to local communities. Projects that do not involve distribution of food items in refugee settlements must include the host community to a minimum of thirty percent. In practice, this means that thirty percent of beneficiaries would be the host community and seventy percent refugees. It is important that the host community is involved as the loss of trees has an impact on them too. This also helps to build social cohesion between the two populations. Since the tree cover in the settlements has been reduced, the refugees need to walk further to obtain firewood, sometimes crossing into the land belonging to host communities. The latter also uses firewood for cooking and is not always inclined to share the valuable resource as will be discussed further in the next sections. The refugee settlements have been experiencing environmental pressures in the form of droughts, low soil fertility and strong winds. These pressures can be linked to the reduction in tree cover. ICRAF established the project to help environmental recovery in the settlements and so that refugees can continue producing sufficient yields to sustain their households.

> *"Establishing agroforestry on land that currently has low tree cover has been identified as one of the most promising strategies to raise food production without additional deforestation"* [24].

### 1.2. Organisations Supporting Agroforestry in Uganda

ICRAF is a research organisation focused on agroforestry. This particular project in the refugee settlements of Arua district is about alleviating the social and environmental stress that tree cutting has stimulated in the refugee settlements. The experts at ICRAF ensured that the correct tree species were selected for each location. They also hired refugees and hosts to work with them, providing training and employment opportunities.

This project can also support with issues surrounding land tenure. Refugees can access land but for many it might be that the tenure is insecure, the land size is small and economically marginal. This is perhaps expected in areas where the population is exerting pressure on local resources [25]. Thus, land space should be maximised for refugees to be able to make the best use of the land they can access. It can also check the practice of deforestation because if refugees manage to produce enough on their allocated land, they will have fewer incentives to clear the forest for agricultural production [26].

Adaptive systems of farming can help conserve natural resources; this can be achieved when higher crop yields are obtained and sustained, and thus, sustainable agriculture practices are performed within a long-term view to promote self-reliance. The successful use of sustainable practices can form a precedent and allow governments to see its benefit and allow for further support and land allocations. Thus, creating a shift in policy.

Agroforestry also brings a decrease in environmental damage post-operation and the local communities could have something to take on over if/when the settlements are gone. Nevertheless, the opportunities will overflow to the local populations even during the life of the refugee settlement as they also adopt sound agricultural practices [23].

In reality, refugees live in settlements for, on average, seventeen years and often much longer and the influx of refugees has not stopped increasing, especially in Uganda [27]. Many refugees find themselves dependent on food aid for sustenance and encounter challenges to rebuild a livelihood. In addition, refugees also tend to place a strain on the host country causing economic, environmental, and security issues on a national basis and also more local challenges impact and local communities [28].

Thus, ICRAF's project also aimed to get the refugee community involved in the environmental issues that surround them. Uganda is a country where agroforestry is very strong in practice and promoted in policies [29,30]. Uganda has committed to the Bonn Challenge and the government has pledged to restore around 2.5 million ha of degraded land by 2030 [31]. The IUCN identified agroforestry, woodlots and farmer-managed natural regeneration as the most cost-effective means of achieving this target [31]. National policies include Forest Policy of 2001 and the National Forest Plan of 2013 and National Forest and Tree Planting Act of 2003. These policies recognise the capacity of agroforestry to provide sustainable livelihoods to farmers with an increase in their productivity and consequently their income [32]. However, certain regions of the country are currently at risk of famines because of droughts and poverty [33]. When communities are at risk, the provision of food aid is usually the main response, similar to the refugee response. However, this only serves to alleviate the symptoms rather than the cause of food insecurity [34]; especially for the poorest, for whom self-reliance and income generating opportunities are quasi-inexistent and being in this category usually means being more exposed to livelihood threats [35]. Thus, the design of programmes should reflect this reality and take into consideration prevention methods such as agroforestry.

Coe et al. state that agroforestry practices are increasingly being encouraged to enhance food security, biodiversity conservation, and diverse ecosystem services [4]. According to the Food and Agriculture Organisation of the United Nations (FAO), 1.5 billion people worldwide benefit from trees in a direct or indirect manner. Trees are an important source of food and can play a critical role in communities who suffer from food insecurity and malnutrition [16]. People have access to trees in forests but trees on farms are becoming a more common practice as farmers realise the opportunities. In fact, agroforestry is one of these practices that are used by farmers and agencies to tackle food insecurity [13,14].

### 1.3. Study Aims and Objectives

Although the economic, environmental and social benefits of agroforestry have been demonstrated in many studies as described above, there is very little research to date on the impact of agroforestry in refugee camps. This study aimed to address this knowledge gap by evaluating the impact of two agroforestry schemes in the Arua district of Uganda provided by ICRAF. The main objectives were (1) to understand how the agroforestry

schemes were implemented in the refugee settlements; (2) to establish how agroforestry provides benefits for the participants and the environment; (3) to identify the opportunities and challenges to wider participation.

## 2. Materials and Methods

### 2.1. Study Area

Northern Uganda was ravaged by war during the 1980s and for decades afterwards by rebel groups. This conflict affected the local population in the districts, causing them to flee to other districts and even neighbouring countries. Uganda is now relatively stable, both in the security and political sense and offers refuge with supportive policies for refugees that support them to settle in the country. The total number of refugees in Uganda is around 1.4 million in 28 refugee settlements. The majority of refugees, 1,061,892, are from South Sudan, followed by the Democratic Republic of Congo (DRC), Burundi, and Somalia [36].

Arua district is located in the north-western part of Uganda and is bordered by the districts of Maracha, Koboko, Yumbe and Moyo to the north, Nebbi and Zombo to the south and Adjumani and Amuru on the eastern side. The west of the Arua district borders the DRC and is also close to border with South Sudan which lies to the north of Yumbe district (Figure 1). The proximity to these two countries is one of the reasons why refugee settlements are prominent in northern Uganda. Indeed, the DRC and South Sudan are countries that are frequently subjected to civil unrest and conflicts, which drive the affected population to cross over to Uganda to seek refuge.

LOCATION OF ARUA DISTRICT IN UGANDA

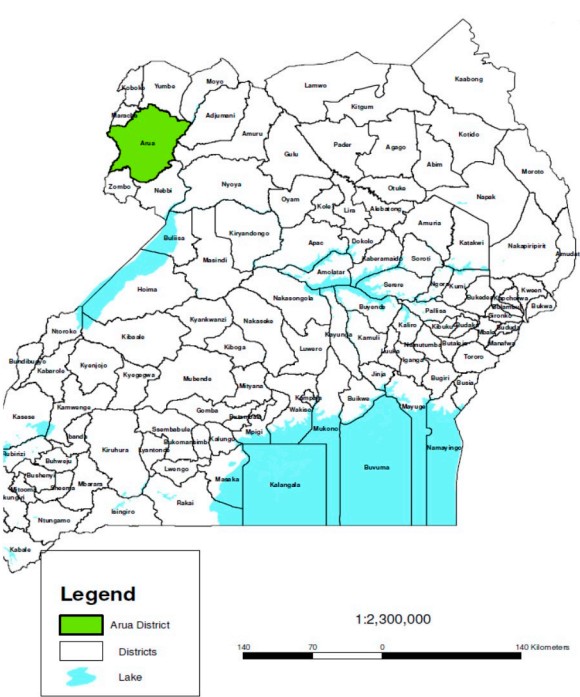

**Figure 1.** Location of the study area in Uganda [37].

Arua district has a total population of 846,491, covers an area of 3236 km$^2$ and hosts 263,018 refugees, primarily from South Sudan in the two largest settlements called Rhino Camp and Imvepi [36] (see Figure 2). Thus, refugees account for twenty four percent of the total district population. Imvepi has 127,084 refugees in an area of around 53 km$^2$ divided into 3 zones, Kaowa, Siripi, and Nawu. Two rivers form the borders of the settlement, the River Ore flows between Arua and Yumbe District and forms the northern border and River Anyau forms the southern border. The altitude ranges from 630 m

in the Rift Valley to 788 m. Rhino Camp has 123,243 registered refugees and covers an area of 85 km². The Rhino Camp project area lies inside the shallow Albertine rift. The Anyau River to the north and the Arua-Rhino Camp main road to the south form the borders of the settlement. Rhino Camp has six zones: Eden, Siripi, Ofua, Tika, Ocea, Tika and Odobu. The settlements are governed by the Office of the Prime Minister (OPM) for refugee operations and monitored by UNHCR. OPM oversees issues regarding land, including allocations, legal matters, and enforces laws. The role of UNHCR is to monitor and support programmes implemented in the settlements. There are several agencies, local and international, that work within the settlements in various sectors, including Water, Sanitation and Hygiene (WASH), Safe Access to Fuel and Energy (SAFE), Livelihoods, Environment and Education.

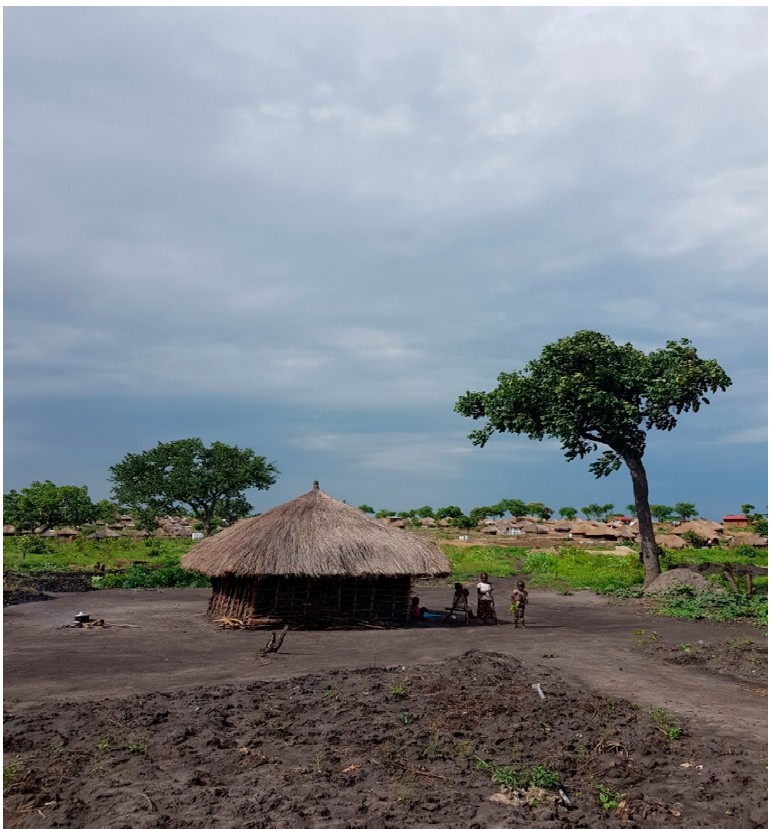

**Figure 2.** Rhino Camp, Refugee household (E. Grosrenaud, 2018).

The data were recorded within a month of the start of the first planting season, which usually begins after the rainfalls between March and April. The rainy season in Arua can last until May and, if lucky, until June. The second planting season takes place in October to November or again when the rain starts to fall until December. The average rainfall is 1400 mm, and the climate is usually dry when out of the rainy seasons.

*2.2. Data Collection and Analysis*

Semi-structured interviews were conducted in the refugee settlements of Imvepi and Rhino camp. The respondents were all beneficiaries from the ICRAF agroforestry project, which means that every person interviewed had received free tree seedlings donated by ICRAF to plant around their homes or on their lands. In total, forty beneficiaries were selected by the community workers, twenty from each settlement. The selection was dependent on the time participants had available for the interview. There were 23 men and 17 women interviewed in both settlements and all interviews apart from three were translated from the local languages into English by the community worker or another member of the community. The age of the participants ranged from 18 to 83 years and

included three members of the host community. Interviews with community workers and tree nursery technicians also took place to collect information about tree species and distribution.

The data were analysed using Excel and SPSS and NVIVO for qualitative results. Pseudonyms were used to preserve the anonymity of the respondents as some of the answers given are sensitive.

The study methods were approved by Coventry University's ethics approval system (project identification code P45146). Participants were provided with a participant information sheet and written informed consent was obtained prior to the interviews taking place.

## 3. Results

Agroforestry considers three interacting components, livestock, agriculture and trees, where the latter is mixed with one or both other components. In refugee settlements, there are different sets of challenges encountered, especially with agroforestry as it is relatively uncommon for refugees to be planting trees on the limited amount of land they can access. Thus, the research looked separately at each of the three aspects: livestock, agriculture, and trees, examining how each of these can benefit refugees and identifying the challenges faced by refugees in these areas. Summaries of benefits and challenges identified by participants are presented in Figures 3, 4 and 6.

### 3.1. Livestock as an Investment

Almost half of respondents did not own any livestock, this was mainly because of a lack of capital, but also a lack of space. Of the respondents with livestock, most kept chickens. Out of all the respondents, over a third had lost poultry to disease. In such a remote location, access to medicine or a veterinarian is limited. Respondents also indicated that even if they had access, they would not have the funds to procure medicine for their animals. For example, Abdullah, 24, used to have 35 chickens but now only has three remaining and feels like he lost significant potential income and there is nothing he could do about it. He described how he came with his poultry from South Sudan:

> "I put it under my arm, and I walked. I knew I was not going to have anything waiting for me there, so I took what I could."

Selling poultry is the livelihood of Fiza, 18 years old; she still lives with her parents but takes on many responsibilities in her household and even encouraged her family to plant trees. She sells a chicken between UGX 10,000 (GBP 2.44) to UGX 20,000 (GBP 4.88). She is satisfied about her source of income but finds it hard to find customers. In addition, she is concerned about finding enough food for her fifty plus chickens and ducks. She states:

> "It is difficult to grow enough food for the family and even more to feed the animals, but we try because the income from selling chicken is good."

Out of all her economic activities, which include selling some of the harvest or rations, selling chicken is the most profitable for Fiza. Many other respondents also complained about not having sufficient feed for their animals. In this sense, it is even harder to keep larger livestock species. Only a third of the individuals interviewed said they possessed cattle, and these were all members of the host community rather than refugees. However, even the host community members were complaining of access to fodder during the dry season as they find it difficult to feed their animals during that time of the year. They would move their livestock for grazing and sometimes let them roam freely to graze unattended. They were keen on tree planting because it would provide them with a sustainable source of fodder during the dry season. Refugees were for the most part keeping goats rather than sheep as these are not very common in the region. Goat keeping also comes with its challenges, including lack of space and fodder, no access to medicine and impact on relationships with neighbours. In fact, refugees who did not own any livestock, when cash was not the issue, said that it was the case because they did not

have enough space to keep and feed animals. Some of them mentioned that they do not want to cause any problems within their communities because animals can destroy crops. Indeed, refugees reticently mentioned that animals had destroyed crops and that the animals belonged to host community members. If it happens during the day and they catch the animal, the matter goes on to the chairman of the village and the refugee will somehow be compensated for their lost crops. However, if it happens during the night there is no way to prove the cause and they must incur their loss. The only solution for the refugees is to put up fences because as Umar mentions:

> *"We do not want to fight with host communities because they help us in many ways, we want to get along, so fencing would be the solution to protect our crops".*

Hinata would also like to have a fence to keep her goats in. She shares the same concerns as the others, but wood for fencing is too scarce and too expensive. She needs to rent land in the host community for her goat to graze on, it is far away and takes up a lot of her time. She rents the land at UGX 50,000 (GBP 11.00) per year for an acre, where she uses half for grazing and half for agriculture. She is forced to rent the land for her goat because she had used all of her allocated land to build her house. She has planted some crops around it, but it is not sufficient space to grow adequate food for her household of seven. Hinata's aim is to grow enough food to sell because she managed to get a small income by selling part of her rations. She then used the cash to buy the goat and claimed that if she had eaten all her rations, she would have not benefited from owning livestock today. She mentioned that she does not use all her ration but divides it for selling so that in the future she can buy more livestock. However, she also faces challenges within her household, she says:

> *"A woman can manage to save to buy a goat, but the husband uses the money to buy cigarettes and alcohol. A woman can save 100 shillings (5p) each time she gets something and then can manage to buy something with it".*

She is hoping to buy a second goat because when she invests her cash into something concrete her husband cannot use the money to drink. In addition, she will increase her income if she manages to rear goats. Indeed, respondents who had goats mentioned that if they need emergency cash, they would sell their livestock to buy medicine or pay for school fees. One host community member, Aloysius, 53, sold twenty cows to send his daughter to university. He stated that livestock is the venture that provides the most when a large sum is needed. (see Figure 3)

| Challenges | Opportunities |
|---|---|
| • Diseases - especially amongst poultry<br>• No access to veterinary and medication<br>• Lack of space/land<br>• Need to rent land for grazing<br>• Lack of fodder especially during the dry season<br>• Animals destroy crops and cause friction | • Provide additional income<br>• Most profitable agroforestry venture<br>• Emergency cash |

**Figure 3.** Summary of key challenges to and opportunities for silvopastoral systems in refugee camps identified by the participants.

### 3.2. Agriculture in Challenging Conditions

From the forty participants, almost all received their land as soon as they arrived to enable them to construct a house and begin agricultural production. The ones that did not perform agriculture from the start were either waiting for a new planting season, doing a training course or were unable to because of illness. Drought and pests were the most common factors that affect crops. Pests included the African armyworm (*Spodoptera exempta*) which mostly destroys maize, one refugee reported that in 2016 most of his maize was gone, because of the worms, he harvested almost nothing. He mentioned that it was

because it rained heavily that the armyworms arrived, because during the dry season they are not present. The African maize stalkborer *(Busseola fusca)* is another common pest found in the settlement as well as Mound-building termites *(Isoptera termitidae)* and Grasshoppers *(Acanthacris)*. Respondents find it extremely difficult to fight pests; there are no pesticides available and even if they could afford to purchase some, they would still need to purchase a pump for spraying, making it an expensive venture. To mitigate this circumstance, ICRAF included *Neem* on the list of trees to plant because it can be an effective insect repellent.

Regarding drought, the settlements endure long dry spells where rain is rare and plants wilt because of the heat. Watering is a difficult option because the distance to the boreholes is often too far. In addition, waiting times to collect water at boreholes make it impossible to collect enough for the household and for watering crops, even on small parcels of land. In contrast to drought the settlements encounters periods of heavy where crops also suffer and are sometimes destroyed by the weight of the rain.

Further challenges that respondents faced included the lack of appropriate tools and that the ground is difficult to dig because there are very large stones that they must take out. Nevertheless, all respondents were content to have space to farm because it helps them start over and produce food after a few months. Munir, 48, said that agriculture helps him not to think about his problems because he had something to do from the start, and did not sit somewhere doing nothing and thinking negatively.

*"I am happy because my children will get something to eat".*

Amina, a mother of nine said that it helps her children when she plants maize because she can feed them with porridge. Another mother said that planting maize is the most important because she can get regular income, she sells 10 cups of maize for UGX 4000 (GBP 0.80). Out of all the respondents, nearly half were selling part of their rations, harvest, or trees to obtain some cash or income. The small amount they get is often used to buy soap (UGX 1000/GBP 0.20) or salt (UGX 500/GBP 0.10). Women put a particular emphasis on how access to soap is really difficult.

They also use the income to diversify their diet or barter some of their harvest with other refugees or host community. One of them explains that when the organisation in charge of the agricultural sector, when they had just settled, gave out seeds, they gave everyone different types. However, many wanted to receive tomato seeds because *"it is the crop with the highest profit."* Five or four tomatoes are sold at around UGX 2000 (GBP 0.40). Seeds are difficult to get. If they were not distributed by organisations, refugees would have almost nothing to plant. Another valuable crop is sesame, or simsim in the local language. Reshma sells two basins of it for about UGX 100,000 (GBP 21.00); this is about half of her harvest and she uses it to pay for the fee for her children to go to school or buy washing soap for clothes. Agriculture provides regular income to many, but others are not able to get income from it because they do not have enough land, or they harvest low yields because the soil is poor in nutrients.

Within the refugee community, land sizes varied from 225 m$^2$ to 1200 m$^2$, but on average a plot size was 900 m$^2$ per household. Those who managed to sell enough started renting land from the host communities as mentioned earlier—this is a common practice. Some even mentioned receiving land for free from host communities. In fact, two out of three members of the host community interviewed stated that they gave land for free to refugees and that they have a good relationship and barter with each other.

*"I give them land to plant simsim (sesame) because it is delicious, then I barter with them to get some".*

Syed is a successful refugee because he manages a large area of land and sells the produce for profit to the host community and fellow refugees. He rents land from the host community and then hires labour from them as well, making it a profitable venture because he manages to plant several crops on a large parcel of land, about 10,000 m$^2$. Without capital, labour is usually difficult to find, workers are usually paid UGX 3000

per day (8 a.m.–1 p.m.). Even the host community members were finding it difficult to find labourers but were also hiring refugees to fill out the labour gap. The majority have good relationships with host communities. However, this is not the case for everyone. On several occasions, refugees have reported that host community's cattle are left to graze freely and sometimes come into their parcel of land and destroy their crops. Furthermore, a few refugee women mentioned that they were struck or shouted at because they went to collect firewood in the host community area. One of them expresses:

> *"They are tired of us taking their firewood, I can understand but we have no choice, there is no firewood left in the settlement, as you can see there are very few trees remaining. People cut them all to build their houses and to cook when they first arrived, and this is not the first time we came here."*

Firewood is one of the reasons she decided to plant trees. Getting firewood is also part of the income opportunities as illustrated in Figure 4, and Figure 5 summarises the different challenges faced by refugees and host communities living in the settlements.

| Opportunities | Challenges | |
|---|---|---|
| Diversify diet | Drought | No labour |
| Regular income | Access to tools | No pesticides |
| Work/Activity | Pests and diseases | Stones |
| Everyday food | Small land size | No pump for pesticides |
| | Destruction of crops by animals | Lack of knowledge |
| | Limited access to quality seeds | Low harvest |
| | Low soil fertility | Heavy rains |
| | Wilting of crops | Long distance to bore holes |

**Figure 4.** Agriculture in challenging conditions, a summary of key opportunities and challenges identified by the participants taking in the agroforestry projects.

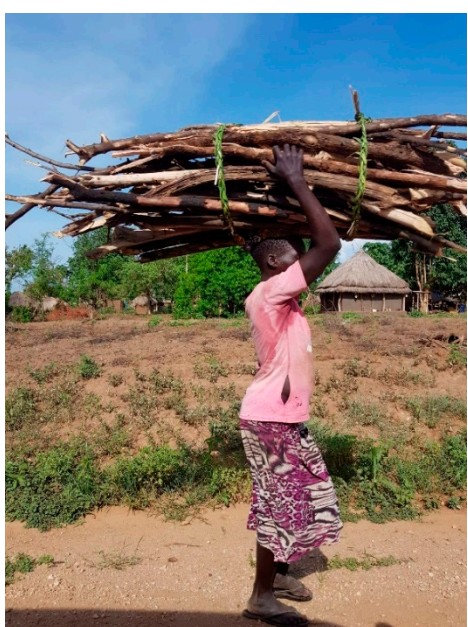

**Figure 5.** Woman carrying firewood for selling or cooking in refugee settlement in northern Uganda (E. Grosrenaud, 2018).

*3.3. Trees*

Access to firewood is a challenge and is exacerbated by the large influx of refugees coming to settle in an area at the same time. Putting immense pressure on the environment.

Many need to construct new homes and cook three times a day, making most of the trees inside their communities disappear. Thus, they are pushed to go into the host community areas where some reported positive experiences such as the locals helping to gather firewood, but others reported negative interactions such as those mentioned above. Others barter some of their food in exchange for firewood and those who have the means purchase it from the host communities. Apart from incidences with firewood and destruction of crops and trees by animals, the relationship between hosts and refugees is rather positive and valued. Mohammed explains why it is important to plant trees to maintain this relationship:

> *"We came in the past and went back to our country, but now we are here again, and we have cut all the trees. What if we come go back to Sudan and come back again? We cannot keep cutting trees without planting new ones. Not just for our benefit but also for the host communities. We do not want them to think that we take advantage of the place. We want to leave them happy. We have a good relationship and we want to keep it positive, planting trees is for everyone."*

Suliman adds on to this by stating:

> *"I do not mind planting trees, even if I go back to my country, I know that someone here will enjoy the trees I am planting, and I can leave all the trees for them. I will not even cut them to sell them if I am going back home because I want them to remember us and be happy that we stayed here. We never know if we have to come back and they should be happy to see is us, not angry with us because we have cut all their trees."*

Firewood was the reason for planting trees for more than half of the refugees interviewed and the third most important one. The most common reason was for shade at 72.5%. Indeed, at the tree nursery in the refugee settlement, neem *(Azadirachta indica)* was not only popular because it is commonly known as an antimalarial but also because it is planted for shade in the compound and is thus in very high demand.

> *"There are no big trees around, comments the nursery technician, so we encourage them to plant certain trees for shade."*

Refugees really suffer from not having shade to rest from the severe sunshine during the day, especially at the sun's peak hours. Some go sit with their neighbours because they have no trees on their land. During the interview, Akifa mentions that:

> *"This tree, I planted it when I first arrived and now, we are using it to speak under its shade, if this tree was not there do you see how we would suffer, the sunshine is too much."*

The second most important factor for planting trees was for poles, mainly for house construction or repairs, 60% of refugees planted trees in the hope of getting poles. There is a high demand for poles, this is the reason ICRAF provides trees that are fast growing; for example, with Melia *(Melia volkensii)*, it is possible to get a pole within 7 months. There are several fast-growing trees that are being distributed, including Leucaena (*Leucaena leucocephala)* and Calliandra *(Calliandra calothyrsus).* Refugees can start cutting branches for firewood without cutting down the tree and let it grow to provide shade and fodder and improve the soil conditions. Refugees are not encouraged to cut down the trees they plant but make use of the services they provide. However, the use and demand for poles is so high that forbidding them to cut the trees they planted is impossible; instead, they are encouraged to replant more trees or take care of the ones that are sprouting. Prices for poles range from UGX 500 (GBP 0.10) to UGX 10,000 (GBP 2.44). This is because access to market is difficult and some prefer to sell for quick cash. An example is Arya; she was selling poles at UGX 500 because it is difficult to get money, thus, she is welcoming any amount she can get. Others said that they have given poles for free to their neighbours. "I had trees, so I cut one and gave it to my neighbour who needed it." Murtaza also mentioned that

> *"Everyone should plant trees, not everyone wants to do it because they think that they will go back home soon, we could all go back in the next few months, but what if we are*

*stuck here for many years? People do not think that far, they only think about today, and tomorrow they will suffer from having nothing. But they will realise that they should have planted trees because they will see that the ones who did will not suffer like them."*

Timber was also a valuable asset to have but only 10% of refugees were interested in selling timber as it usually takes much longer to have quality wood. This is compared to the host community where 100% of them said that timber was an important motivation for them to plant trees. The prices for timber are much higher than for poles and at the time the market price was UGX 50,000 for one mature tree.

From the question posed on whether planting trees will help improve their living conditions, 100% of refugees and host community responded positively. In fact, economically, 25% of respondents believe that they can get an income from selling poles or fruits at the market or to their neighbours and host community. Many wanted to get more fruit trees but because the demand is so high there are only four per compound allowed—they get two Papaya (*Carica papaya*) trees and two jackfruits (*Artocarpus heterophyllus*). Omer stated that they want a more balanced diet as the food aid they receive is almost the same and they would like to receive better nutrition. Thus, for him having fruit trees is important for his children. Trees are also beneficial for children in other ways. For example, Munirah mentions that she can sell a pole to get medication for her children, because some drugs are not in the health centre, they need to buy them at a different clinic, and they are not free. Another mother mentioned that, for her, selling the poles will be used for paying school fees for the smaller children in primary school, as the fee usually goes around UGX 3000 per trimester, she can sell a few poles and pay the fee. Schools supported by the government are free, but parents feel like they are overcrowded and that their children do not learn as well as in private schools. Thus, they try hard to pay for the school fees for their children. Some even manage to pay higher fees for their children in secondary school. Planting trees becomes a source of cash, because when they need it, they can cut a tree and use it for school fees or buying treatments. One member of the host community, Peter, mentioned that he is planting trees as part of his retirement plan, because when they grow there is not much maintenance he needs to perform. The only downside to his plan would be that other refugees can sometimes fell trees or cut off branches without the knowledge of the owner. However, Peter seems to be determined with his plan:

*"Right now, it is time consuming to plant trees, but I do it because later on I do not have to put in effort, if I need money, I can ask someone to come and cut them and I just sell them directly. Because when I am old, I will not be able to do difficult labour like I do now. I am planting 5,000 trees and there is a lot of work involved and I need to pay workers to help me dig and remove weeds, but I know it is worth it for later on".*

With the support of ICRAF, he received the seedlings for free but he will need to pay for the labour and the maintenance of the trees. Such a large number of trees can be challenging to take care of but ICRAF would not have given them the trees if they did not have the capacity to see them grow into maturity. There are always losses, but they should be minimal.

Furthermore, he mentioned that he is planting such a large number of trees because he wants to pay school fees for his children in the future. He is hoping to sell a tree for timber at UGX 50,000 and even send his children to university. Out of the host communities interviewed, they were planting trees on a much larger scale than refugees. This is because they had more land and more resources available. One community worker stated that they do not give trees if people cannot take care of them because they will just die but if they have the resources available, they can receive as many as they desire. The host community noticed the degradation on the environment and offered land for refugees to plant trees. Their reasoning behind giving away land is that they have sufficient land for themselves and they cannot manage everything. In addition, the host community realises the effects that tree felling has on the environment and they are also being affected by droughts. So, instead of having unused land, they can help the refugees be more sustainable and at the

same time conserve the environment on their lands and mitigate risks of climate change. Several refugees received free land and have different arrangements with the landowner. Some said that they will use the firewood together and if they sell any trees, they will split the profits with the landowner. Others said that they are planting the trees so that they do not need to go collect firewood and will leave the trees to the landowner when they go back to Sudan. Moreover, others planted trees specifically to improve the soil fertility and will give some of the harvest to the landowner because he is their friend.

*"If he gives me land for free, I will also give him something for free when I have, we are the same. He never asked for anything in return. He had land he was not using because it is too far from his home, but he lent it to us, so we can use. So later on, I also want to do something nice for him."*

There is a potential to limit issues between the host communities and refugees by making sure that firewood is collected in a more sustainable way and livestock is kept out of plantations. The tree species that were distributed have the potential to resolve this; they are adapted to the local area which is semi-arid and rocky. Tree species distributed by ICRAF are shown in Table 1. Trees here generally withstand low moisture and high temperatures, for example, *Albizia gummifera, Senna siamea* and Terminalia (*Terminalia brownii)* are species that can typically withstand those conditions. Refugees also planted trees because they wanted to attract more rain—22.5% said that it will help with rain if they planted trees, so they will have better crops. Another important factor for refugees is the provision of windbreaks—35% said that it was part of the reason they decided to plant trees. One woman has a very poignant explanation:

**Table 1.** Tree species produced in the International Centre for Research on Agroforestry (ICRAF) nursery at Imvepi and Rhino Camp.

| Plant Species | Local Name | Indigenous/ Exotic | Function/Uses | Preference | Growth |
|---|---|---|---|---|---|
| *Khaya grandifoliola* | Khaya | Indigenous | Fodder/Antimalarial/Timber/ Planted near water borders | Host community | Slow/20 years |
| *Leucaena leucocephala* | Leucaena | Exotic | Fodder/Firewood | Refugees | Fast/1 year |
| *Calliandra calothyrsus* | Calliandra | Exotic | Fuelwood, Fodder, Fibre, Honey, Shade, Erosion Control, Soil Improvement, Nitrogen Fixing/Ornamental | Refugees | Fast/1 year |
| *Artocarpus heterophyllus* | Jackfruit/Fenne | Exotic | Nutrition/Shade/Fodder/Income | Refugees | Medium/3–4 years |
| *Moringa oleifera* | Moringa | Exotic | Nutrition/Medicine/Drought Resistant/Erosion Control/Fencing | Refugees | Fast/2–3 months |
| *Tamarindus indica* | Tamarind | Indigenous | Nutrition/Medicine/Ornamental | Host community | Slow/15 years |
| *Carica papaya* | Papaya/Pawpaw | Exotic | Nutrition/Income | Refugees | Medium/3–4 years |
| *Balanite Aegyptiaca* | Desert Date | Indigenous | Nutrition/Wine/Nitrogen Fixing/Fodder/Fencing | Host community | Medium/5–8 years |
| *Afzelia africana* | African Mahogany | Indigenous | Timber/Soil Conservation | Host community | Slow/25 years |
| *Albizia gummifera* | Peacock Flower | Indigenous | Nitrogen fixing/Soil Conservation | Host community | Fast/1–2 years |
| *Senna siamea* | Cassia | Exotic | Timber/Poles/Mulching/ Intercropping | Refugees | Fast/1 year |
| *Azadirachta indica* | Neem | Exotic | Antimalarial/Shade/ Drought resistant | Refugees/Host community | Fast/1–2 years |
| *Combretum molle* | Combretum | Indigenous | Firewood/Fodder | Refugees | Fast/1 year |
| *Melia volkensii* | Melia | Indigenous | Firewood/Poles/Timber/ Mulching/Intercropping | Refugees | Fast/7 months |
| *Terminalia brownii* | Terminalia | Indigenous | Windbreak/Shade/Mulch/ Intercropping/Drought resistant | Host community | Slow/15 years |
| *Vitex doniana* | Vitex | Indigenous | Nutrition | Host community/Refugees | Medium/3 years |

*"My house was destroyed by the wind, you can see here I did not have the money to reconstruct it. This was done by the wind, so I feel like it is important to plant trees around the house to reduce the wind from destroying our houses. I lost my daughter because of that, she was inside the house and the bricks collapsed on her. We already suffered from coming here and now we have to suffer again because our environment is not safe."*

The other reasons planting trees included diet, with Moringa *(Moringa oleifera)* being a favourite for its leaves, to perform apiculture, change the climate, make new furniture, for better oxygen, as ornaments to beautify the settlements and as a landmark or boundary. (see Figure 6)

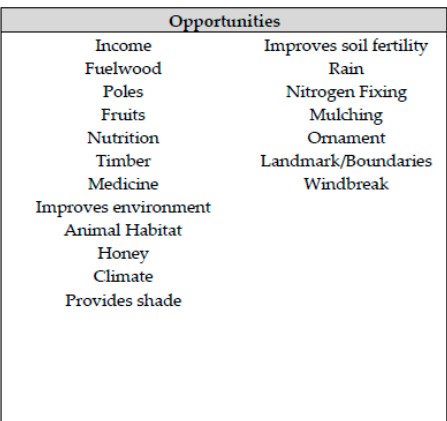

| Opportunities | | Challenges |
| --- | --- | --- |
| Income | Improves soil fertility | Drought |
| Fuelwood | Rain | Trees destroyed by animals |
| Poles | Nitrogen Fixing | Limited land available |
| Fruits | Mulching | Heat (causes wilting) |
| Nutrition | Ornament | Watering (time taken, difficulty accessing water) |
| Timber | Landmark/Boundaries | Pests |
| Medicine | Windbreak | Lack of Knowledge |
| Improves environment | | No Labour available |
| Animal Habitat | | No Tools |
| Honey | | Climate unsuitable |
| Climate | | Types of trees available |
| Provides shade | | No market access |

**Figure 6.** Opportunities provided by agroforestry trees and challenges to their production as identified by the participants.

## 4. Discussion

From the results, we can see that the relationship between the host community and refugees is dynamic with some having very positive experiences and receiving free land or firewood, whilst others had a more negative experience because of the pressure of human activity on resources. Thus, there is a possibility that planting more trees to relieve the stress on the natural resources can also have a positive effect on the relationships between locals and refugees. In fact, Moore mentions that agroforestry programs were used to reduce antagonism between the host Cameroonian community and Chadian refugees [38]. In this example, many refugees decided to plant trees in the camp but also outside it so as to replace the trees they had cut down for firewood and to "heal the environment" as they put it. Furthermore, trees can provide more fodder, which was also a cause of conflict between people, when animals are eating crops because grazing lands are scarce during the dry season. At the same time, it can resolve the issue of not being able to have more livestock because of lack of fodder. Whilst many of the refugees aspired to keep livestock as they saw this as a good investment, a liquid asset, in this study few refugees were able to achieve this. A small number kept chickens, but mainly the host community were able to raise and keep larger animals. Livestock ownership was restricted mainly by lack of capital to invest and the limited amount of land available for the animals, but also to a lesser extent by the cost of caring for the animals. Similar barriers to livestock ownership were identified by Frank [39]. Refugees that managed to invest in livestock did so by selling part of their food aid or harvest, saving over several years to buy a goat, as only rarely would a refugee be able to afford cattle. Agroforestry can also provide emergency cash or regular income by selling, livestock, poles, fruits or agricultural produce when needed. The use for this cash was mainly to pay school fees, medicine, or to buy salt and soap, or diversify diet. Regarding the opportunities on soil fertility and the environment, there is a significant potential. However, at this stage, it is difficult to assess how tree planting will affect crop production, but given the tree selection, there is a relevant chance that soil fertility will increase, and better yields can be obtained with time with the trees that are nitrogen fixing or with the ones that shed leaves that can be used for mulching.

Nduwamungu and Munyanziza encourage reforestation and agroforestry programs to be implemented in refugee camps to rehabilitate the local environment because these methods have the potential to challenge desertification, improve field fertility and amplify the water availability [40]. In addition, trees have the capacity to add nutrients to the soil,

particularly nitrogen that can be used by crops [41]. This can be beneficial in refugee camps where the soil is often of poor quality and can help restore or protect the biodiversity of the area. Drought, wind, and extreme sunshine are challenges faced by refugees in Arua district and most of them planted trees in hope to overcome those.

The key findings of this study clearly suggest that agroforestry programmes in refugee camps are both valuable and feasible. Further research is needed though to address some of the challenges identified in Table 2. These are questions for both the natural and social sciences. Field research is still needed to assess tree suitability for different environments and locations and to provide a wider range of species and varieties to meet beneficiaries needs. For example, some native trees are on the verge of extinction and further research on how to conserve native species in an emergency context is urgently needed for their cultural and social role and the local biodiversity. More research is needed to improve methods and practices in refugee tree nurseries as well as in tree planting and production and this information needs to be effectively disseminated. Social and economic aspects are also critical to sustainability and there is much work to do to understand how best to manage agroforestry resources. In this study, some of the beneficiaries successfully worked together to tend their trees; there is a question then of whether community schemes rather than individual effort are the way forward. There are also policy implications to be explored, e.g., who should fund these schemes, and how does land tenure impact their success.

**Table 2.** Key opportunities for and challenges to the wider adoption of agroforestry in Ugandan refugee settlements.

| | Opportunities | | Challenges |
|---|---|---|---|
| Economic Benefits | Diversifies income sources and can provide an additional regular income source, which can be used to pay for education, also provides "an emergency fund" and long-term income from trees. Reduces reliance on aid. | Lack of resources | Difficult to manage plot without tools and tending trees is labour intensive. Limited land available for agroforestry in settlements. Lack of suitable pest control Availability of locally adapted trees. |
| Environmental Benefits | Multifunctional landscapes that include trees, crops and livestock, having positive impacts on a wide range of ecosystem services including provisioning, regulating, supporting and cultural services. As well as providing additional resources of food and fuel, agroforestry can prevent further harm to the environment by reducing deforestation and provide positive benefits in the form of increased biodiversity, soil fertility and water availability. | Limitations of local climate | Limitations of local climate. Heat and drought cause wilting and can kill young trees before they establish. Watering needs time and effort and depends on proximity of water source. |
| | | Lack of Knowledge | How to deal with tree pests and diseases, about the local adapted trees, how to deal with local conditions, types of trees available. Local conditions. |
| Social Benefits | Fosters cooperation between local community and refugees. Promotes social cohesion. | Other challenges | Trees destroyed by animals. No market access for products. Lack of funding and staff for agroforestry organisations. |

## 5. Conclusions and Recommendations

The refugee crisis in Uganda has substantially increased demand for both fuelwood and poles for construction, resulting in severe deforestation. However, efforts to plant trees are unable to match the rate of tree cutting in the refugee and host communities, despite the fast-growing trees that have been planted. Unless efforts to increase the rate of tree planting and provision of alternative sources of building materials and energy are successful, deforestation and land degradation will likely be increased. These efforts

can be sponsored by UNCHR as they are responsible for settling refugees. The UN must take on the responsibility of conserving the environment in refugee settlements in the same manner they are responsible for the welfare and wellbeing of the refugees. Tree regeneration using conventional planting of seedlings has limitations such as difficulty of raising seedlings and delivering them to farmers for planting, and potentially limited survival rates after planting-out. At settlement level, there is a need to map which areas are fit for providing a supply of wood and to determine the appropriate tree species to be planted. There are also huge opportunities for tree growing among host communities which, if matured, can potentially off-set wood requirements in refugee settlements and moreover provide income to households. Understanding the environmental aspects of humanitarian interventions requires understanding the whole context, which links the environment to the daily life of the refugee and host communities dwelling in a designated geographical space—Arua district in this case. Any intervention to address the environmental concerns (particularly on forest and agroforestry issues) should recognise and also consider the dynamics happening in the landscapes. The dynamics could be due to new practices, new knowledge and new habits that become part of the landscape and hence influence the way natural resources are used and or managed. This is quite common in settings where there are influxes of people from other socioeconomic and cultural backgrounds, like the one happening in Arua district. Such influxes can even ignite new behaviour or concerns from the host communities due to issues of ownership, tenure and other local environmental values which the refugee communities may fail to recognise due to limited knowledge about their new living environments. It is also important to recognise that not all interventions could fit into the contexts of the different communities, specifically the refugee and host communities. ICRAF deployed the "options by context" approach which emphasises the need to choose and design interventions as per the socio-cultural contexts of the communities. This was done by selecting trees that are suitable for both the host community and the refugee population and to attempt to alleviate the pressure on the communities to gather firewood.

Furthermore, refugees should be active participants in preserving the environment in the settlement and larger sensitisation programmes are required to reach out the whole refugee population. Although this project was on a voluntary basis, every refugee should be required to attend meetings on the environment and how to sustainably source wood and to understand the opportunities that planting trees can provide them in the short and long run.

Several refugees understood that agroforestry was not only beneficial for themselves but also to keep a good relationship with the host community. Thus, it would be beneficial that community workers underline that fact during sensitisation meetings. Moreover, stricter rules on tree cutting should be applied but by providing alternatives to refugees. For example, instead of allowing refugees to cut trees on their land to construct houses, they should be provided with sustainably sourced poles on arrival. Moreover, since the host communities are starting to plant on a large scale, it is possible for the wood to be procured locally. This will help local populations to get an income and a market to sell their timber. Procuring poles to refugees would mean that less funds are needed to restore the environment and better relationships are kept with host communities. Dealing with the root cause rather than the effects. Regarding firewood, better alternatives also need to be implemented, several other camps have a briquette factory that are subsidised to make them affordable for refugees. Finally, when the mapping of the settlement is carried out, the environment should be taken into consideration and the areas where refugees can source fuelwood should be designated. Refugees must cook three times a day and being able to feed a large number of people at the start of an emergency is a great challenge but one that should consider the impact on the environment.

Agroforestry in Uganda has the potential to change the lives of farmers by increasing their livelihoods and protecting them from droughts and other climate related harms. Following the Ugandan Policies on Agroforestry as mentioned earlier can help build sus-

tainable livelihoods of farmers including Ugandans and refugees and provide an increased income to many households.

**Author Contributions:** Conceptualization, methodology, formal analysis, investigation, writing—original draft preparation, E.G.; writing—review and editing, supervision, E.G., A.A.-B., C.A.O., L.T. All authors have read and agreed to the published version of the manuscript.

**Funding:** This research was funded by the Department for International Development under a collaborative project between Deutsche Gesellschaft für Internationale Zusammenarbeit and World Agroforestry Centre on Sustainable Management of Natural Resources in the Refugee Context in Uganda. Open access publication costs funded by the Centre for Agroecology, Water and Resilience, Coventry University.

**Institutional Review Board Statement:** The study was conducted according to the guidelines of the Declaration of Helsinki and approved by the Institutional Ethics Committee of Coventry University, project identification code P45146 on 19 December 2016.

**Informed Consent Statement:** Informed consent was obtained from all subjects involved in the study.

**Data Availability Statement:** The data presented in this study are available on request from the corresponding author. The data are not publicly available due to confidentiality.

**Acknowledgments:** We would like to express our special thanks of gratitude to all the staff and fieldworkers that supported the data collection phase.

**Conflicts of Interest:** The authors declare no conflict of interest.

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
