# Peer review of "Agroforestry: Challenges and Opportunities in Rhino Camp and Imvepi Refugee Settlements of Arua District, Northern Uganda"

_sustainability, doi:10.3390/su13042134_

Round 1

Reviewer 1 Report

Sustainability january 19

This manuscript by Gorsrenaud et al. examines how the practice of agroforestry can fulfill several objectives in areas challenged both by humanitary needs as well as ecological restauration.

Tjis manuscript is very well written and includes an introductory section describing agroforestry for the non-specialist, which is welcomed.

The study area is located in the Arua district of Uganda where two settlements of refugees exist: in Imvepi and Rhino Camps, and where the ICRAF has organized a nursery of important local, multi-tree species. The working hypothesis of this manuscript is stated on line 12: "Establishing agroforestry on land that currently has low tree cover has been identified as one of the most 120 promising strategies to raise food production without additional deforestation" (Mbow et al. 20140).

The data was recorded within a month of the start of the first planting season which usually begins after the rainfalls between March and April. Semi-structured interviews were conducted in the refugee settlements of Imvepi and Rhino camp, with 20 people interviewed in each place, all beneficiaries from the ICRAF agroforestry project. All ethical procedures were followed.

The article describes clearly the difficulties that refugees have to secure their own housing, and then maintaining the food provision for their families and their animals if any. Overall, firewood scarcity is one major problem, and often a source of conflict with the local inhabitants.

Motivation to plant trees vary, because many think they will go back home soon.

I find this manuscript very well written and it has a vivid description of what happens in very complicated areas where resources are scant. In other words, it is here where the concept of “Sustainability” is challenged the most, and not in European cities…

The authors show the current and potential role of agroforestry within this context, so the job is done. In my opinion, the manuscript is ready for publication with two minor comments below.

Minor comments

Lines 168 to 188 seem to be out of context as this section 1.2 is about 1.2. Organizations supporting agroforestry in Uganda. Consider presenting it earlier in the introduction?

Would a list of questions be available for future researchers, at least in an appendix form?

Reviewer 2 Report

This paper addresses an issue that tends to be more significant in the past years as the need for sustainable interventions to protect and help vulnerable population becomes more evident. In this context, I think that this study could further add knowledge on this topic and help governmental and non-governmental organizations to shape their interventions by taking account of each particular context that could alter the efficiency of these interventions.

However, there are still some issues with this manuscript that, in my opinion, the authors need to clarify.

First, in my opinion the abstract seems to be a little too vague, I think that the authors should more clearly outline the actual results of their research.

Second, although the introduction is very comprehensive and the literature review is up to date, the authors should point out what is the gap in knowledge which they address and what is the main aim of their research. This info needs to be added whether at the end of the introduction section or at the beginning of materials and methods section.

Third, the discussion section needs further improvements as the authors should more clearly describe how all their research outcomes relate with current knowledge and evidence (for example livestock as an investment which is mentioned in results section it is not addressed in the discussion topic).
